# Combination of Epithelial Growth Factor Receptor Blockers and CDK4/6 Inhibitor for Nasopharyngeal Carcinoma Treatment

**DOI:** 10.3390/cancers13122954

**Published:** 2021-06-12

**Authors:** Hsin-Pai Li, Chen-Yang Huang, Kar-Wai Lui, Yin-Kai Chao, Chun-Nan Yeh, Li-Yu Lee, Yenlin Huang, Tung-Liang Lin, Yung-Chia Kuo, Mei-Yuan Huang, Yi-Ru Lai, Yuan-Ming Yeh, Hsien-Chi Fan, An-Chi Lin, Jason Chia-Hsun Hsieh, Kai-Ping Chang, Chien-Yu Lin, Hung-Ming Wang, Yu-Sun Chang, Cheng-Lung Hsu

**Affiliations:** 1Department of Microbiology and Immunology, Chang Gung University, Taoyuan 33305, Taiwan; paili@mail.cgu.edu.tw (H.-P.L.); hmy0719@mail.cgu.edu.tw (M.-Y.H.); yi_ru@mail.cgu.edu.tw (Y.-R.L.); ysc@mail.cgu.edu.tw (Y.-S.C.); 2Graduate Institute of Biomedical Sciences, Chang Gung University, Taoyuan 33305, Taiwan; 3Molecular Medicine Research Center, Chang Gung University, Taoyuan 33305, Taiwan; 4Division of Hematology-Oncology, Department of Internal Medicine, Chang Gung Memorial Hospital, Chang Gung University, Taoyuan 33305, Taiwan; b9202070@cgmh.org.tw (C.-Y.H.); ldl2605@adm.cgmh.org.tw (T.-L.L.); 8705024@adm.cgmh.org.tw (Y.-C.K.); r94b47405@cgmh.org.tw (H.-C.F.); b9004135@stmail.cgu.edu.tw (A.-C.L.); wisdom5000@adm.cgmh.org.tw (J.C.-H.H.); whm526@adm.cgmh.org.tw (H.-M.W.); 5Department of Medical Imaging and Intervention, Chang Gung Memorial Hospital, Chang Gung University, Taoyuan 33305, Taiwan; kwlui@cloud.cgmh.org.tw; 6Division of Thoracic and Cardiovascular Surgery, Department of Surgery, Chang Gung Memorial Hospital, Chang Gung University, Taoyuan 33305, Taiwan; chaoyk@adm.cgmh.org.tw; 7Liver Research Center, Department of General Surgery, Chang Gung Memorial Hospital, Chang Gung University, Taoyuan 33305, Taiwan; ycn@cgmh.org.tw; 8Department of Pathology, Chang Gung Memorial Hospital, Chang Gung University, Taoyuan 33305, Taiwan; r22068@adm.cgmh.org.tw (L.-Y.L.); dochempath@cgmh.org.tw (Y.H.); 9Genomic Medicine Core Laboratory, Chang Gung Memorial Hospital, Taoyuan 33305, Taiwan; ymyeh@cgmh.org.tw; 10Department of Otolaryngology-Head and Neck Surgery, Chang Gung Memorial Hospital, Chang Gung University, Taoyuan 33305, Taiwan; changkp@adm.cgmh.org.tw; 11Department of Radiation, Chang Gung Memorial Hospital, Chang Gung University, Taoyuan 33305, Taiwan; qqvirus@adm.cgmh.org.tw; 12School of Medicine, Chang Gung University, Taoyuan 33305, Taiwan

**Keywords:** NPC, PDX, EBV, EGF, EGFR blocker, CDK4/6 inhibitor, RNA sequencing, precision medicine, NPC cell line

## Abstract

**Simple Summary:**

Our findings indicated that the EGF-EGFR pathway was highly activated in very young patients with recurrent or metastatic NPC. High EGFR expression in patients with metastatic NPC resulted in poor clinical outcomes. To examine whether the EGFR pathway serves as a therapeutic target for NPC, NPC patient-derived xenograft (PDX) and NPC cell lines were treated with EGFR inhibitors (EGFRi) and a cell cycle blocker. Either EGFRi or cell cycle blocker treatment alone could reduce NPC cell growth and PDX tumor growth. Furthermore, combination treatment exerted an additive suppression effect on PDX tumor growth. This study provides promising evidence that EGFRi used in combination with a cell cycle blocker may be used to treat patients with NPC.

**Abstract:**

Background: Nasopharyngeal carcinoma (NPC) involves host genetics, environmental and viral factors. In clinical observations, patients of young and old ages were found to have higher recurrence and metastatic rates. Methods: Cytokine array was employed to screen druggable target(s). The candidate target(s) were confirmed through patient-derived xenografts (PDXs) and a new EBV-positive cell line, NPC-B13. Results: Overexpression of epithelial growth factor (EGF) and EGF receptor (EGFR) was detected in young patients than in older patients. The growth of NPC PDX tumors and cell lines was inhibited by EGFR inhibitors (EGFRi) cetuximab and afatinib when used separately or in combination with the cell cycle blocker palbociclib. Western blot analysis of these drug-treated PDXs demonstrated that the blockade of the EGF signaling pathway was associated with a decrease in the p-EGFR level and reduction in PDX tumor size. RNA sequencing results of PDX tumors elucidated that cell cycle-related pathways were suppressed in response to drug treatments. High EGFR expression (IHC score ≥ grade 3) was correlated with poor survival in metastatic patients (*p* = 0.008). Conclusions: Our results provide encouraging preliminary data related to the combination treatment of EGFRi and palbociclib in patients with NPC.

## 1. Introduction

Nasopharyngeal carcinoma (NPC) is a prevalent head and neck tumor in Southeast Asia, including Taiwan [1]. Individual genetic susceptibility, Epstein–Barr virus (EBV) infection, and dietary or chemical carcinogens are the main etiological factors contributing to NPC pathogenesis [2,3,4,5]. The majority of patients with NPC are aged between 40 and 60 years, and NPC has been rarely diagnosed in patients aged below 30 years and above 70 years [6,7]. Adolescents and young adults with cancer are likely to have poor outcomes because of the molecularly distinct tumor signature, predisposition to genetic mutations, and severe clinical manifestations [8,9]. Older patients with cancer often have a poor prognosis due to a more advanced cancer stage at diagnosis, the presence of multiple morbidities, and treatment-related toxicity [10].

Genomic, transcriptomic, bioinformatic, and omics studies have indicated that aberrant signal transduction pathways including epithelial growth factor (EGF) receptor (EGFR), nuclear factor kappa-B (NF-κB) [11], WNT/β-catenin [12], PI3K/Akt/mTOR, p53, PARP, MAPK, STAT, cell cycle pathways, and miRNAs/lncRNAs [13,14] may result in tumor progression and treatment resistance in NPC. EGFR is a tyrosine kinase receptor. Upon binding to the EGF ligand, EGFR undergoes dimerization and autophosphorylation at its C-terminal domain [15]. This, in turn, activates downstream signaling pathways such as RAS/RAF/MEK/ERK, PI3K/AKT, and PLC/PKC [16]. Activation of the EGFR pathway was detected in NPC cell lines with high metastatic potential than in those with low metastatic potential [17]. EGFR was highly expressed in NPC and contributed to increased metastatic risk and poor overall survival [18,19]. Conceivably, inhibitors including the EGFR monoclonal antibody and tyrosine kinase inhibitors [20] that block EGF-EGFR signaling and a cell cycle blocker could impede tumor growth. To test this hypothesis, EGF-EGFR inhibitors and/or cell cycle blockers were used to inhibit NPC growth in our NPC patient-derived xenograft (PDX) mouse model [21] and a recently established NPC PDX-derived cell line. Although EBV is a crucial viral factor in NPC pathogenesis and progression, most of the available isolated NPC cells have lost the EBV genome during long-term passaging, hindering progress in this field [22,23]. Currently, NPC PDX and PDX-derived EBV (+) cells are more effective than EBV (−) cells for mutation identification, drug screening, and drug response prediction because they preserve the EBV genome and can recapitulate the original tumor’s heterogeneous genetic profile [24,25]. Drug tests in NPC PDX animal models and RNA-seq data may reveal that EGF signaling and cell cycle inhibitors effectively suppressed NPC tumor growth.

## 2. Materials and Methods

### 2.1. Drugs

Palbociclib (Ibrance) was purchased from Pfizer Manufacturing Deutschland GmbH (Freiburg, Germany). Afatinib (Giotrief) was purchased from Boehringer Ingelheim Pharma GmbH and Co. KG (Ingelheim am Rhein, Germany) and cetuximab (Erbitux) was procured from Merck Healthcare KGaA (Darmstadt, Germany).

### 2.2. Cell Growth Assay and Animal Studies

NPC cell lines, C666-1 (EBV positive) and HK-1 (EBV negative), were maintained in RPMI containing 10% fetal bovine serum (FBS). A cell growth assay and animal studies were conducted as described previously [24]. All experiments involving laboratory animals followed the guidelines for animal experiments of Chang Gung Memorial Hospital (CGMH) and were approved by the IACUC of CGMH.

### 2.3. Patients with NPC

Blood/plasma samples from patients with NPC were prospectively collected after written informed consent was obtained between January 2007 and December 2015 at CGMH. A total of 502 patients with NPC were determined to be eligible. Among 502 patients, 186 had recurrence or metastasis, and 316 were disease-free for more than five years. This study was approved by the Institutional Review Board of CGMH.

### 2.4. Plasma Cytokine Array Analysis

A cytokine array assay (kit ARY022, R&D systems, Minneapolis, MN, USA) was performed according to the manufacturer’s instructions, and this assay allowed the parallel determination of 102 cytokines in each sample [26]. In brief, patients’ plasma samples (500 μL) were diluted with blocking buffer and incubated overnight with a pre-blocked cytokine array membrane. The membrane was washed and then incubated with biotinylated detection antibodies. Streptavidin-HRP and chemiluminescent detection reagents were used. Signals corresponding to different cytokines were captured using an X-ray film.

### 2.5. Enzyme-Linked Immunosorbent Assay of Cytokines

Human plasma EGF was measured using the Quantikine human EGF enzyme-linked immunosorbent assay (ELISA) kit from R&D Systems (Minneapolis, MN, USA) according to the manufacturer’s instructions and modified as described previously [27].

### 2.6. NPC PDX Establishment

PDX models were generated as described in our previous study [24]. In brief, NPC tumor samples were obtained from patients with local recurrence or metastasis undergoing biopsy or surgical resection. Each sample was immediately cut into smaller pieces of approximately 3–5 mm in diameter and subcutaneously implanted into the flank regions of anesthetized NOD/SCID mice. The xenografts were excised and reimplanted in small pieces into the next passage after the tumors reached 1 to 2 cm in diameter. The nomenclature system for our NPC PDX was NPC PDX-metastasized tissue (abbreviation)-patient number. Accordingly, the NPC PDX-ST1 and PDX-B13 corresponded to NPC PDX-soft tissue and NPC PDX-bone in reference [21].

### 2.7. New NPC-B13 Cell Line Establishment

NPC PDX-B13 (NPC13F5) [21] was cut into pieces, and cells were then dissociated through repetitive pipetting in culture medium. Dissociated cells were pelleted through centrifugation and cultured in Dulbecco’s modified Eagle medium (Nutrient Mixture F-12 (DMEM/F12), (Gibco, Grand Island, NY, USA) supplemented with 1% FBS, 2 μM hydrocortisone (Sigma, St. Louis, MO, USA), 1% N2 supplement (Gibco, Grand Island, NY, USA), 1% insulin–transferrin–selenium (Gibco, Grand Island, NY, USA), 20 ng/mL human EGF (Gibco, Grand Island, NY, USA), 2 mM L-glutamine (Gibco, Grand Island, NY, USA), and 100 U/mL penicillin-streptomycin (Gibco, Grand Island, NY, USA). Although the NPC-B13 cell line was passaged for more than 80 passages in vitro, it did not form a xenograft in NOD/SCID mice. NPC-B13 was under depositing to ATCC processing, PO082527.

### 2.8. Quantitation of EBV DNA in the Xenograft

DNA was extracted from the plasma or tissue as described previously [24]. The method for EBV DNA quantitation was described previously [28]. The relative EBV DNA concentration in the tissue is expressed as EBV DNA copies/β-actin gene copies in the same tested samples.

### 2.9. Bulk RNA Sequencing and Differential Expression Analysis

Total RNA from PDX tissues was extracted using TRIzol (Thermo Fisher Scientific, Waltham, MA, USA) as described previously [21]. Subsequently, the RNA sample was sent for high-throughput sequencing by using the NovaSeq platform and KAPA mRNA HyperPrep kit. Raw FASTQ data were trimmed using Skimmer software with the default parameter to remove reads with adaptor contamination or poor quality [29]. The trimmed reads were then aligned to a human-mouse hybrid reference genome (hg38 and mm10) by using STAR software to identify murine reads [30]. Reads aligned to the murine genome were removed, and reads aligned only to the human genome were used to generate gene counts. Raw gene counts were calculated using STAR software with GTF file version GRCh38.p7 (accession number: GCA_000001405.22). Differential gene expression was examined using R version 4.0 and DESeq2 [31]. Gene ontology (GO) enrichment and pathway analysis were conducted using the cluster Profiler R package [32].

### 2.10. Antibodies

One hundred micrograms of the protein lysate per lane were used for Western blot analysis. Antibodies used in this study were as follows: EGFR (Santa Cruz SC-373746), EGFR-p (Abcam Ab40815), ERK1/2-p (Cell Signaling 9102S), tubulin (Upstate 05829), RB1 (CusaBio PA003948), RB-p (Cell Signaling 9307), E2F1 (Santa Cruz SC-193), CDK4 (Santa Cruz SC-23896), CDK6 (Santa Cruz SC-53638), CCND1 (Santa Cruz SC-8396), and PCNA (Proteintech 10205-2-AP).

### 2.11. Statistical Analysis

Tumor weight data are presented as mean ± SD. The final tumor volume was compared using a two-tailed analysis of variance (ANOVA). Overall survival, which was calculated from the time of recurrence or metastasis to death, was examined by plotting Kaplan–Meier curves and compared using the log-rank test. In all analyses, *p*-values were two-tailed, and data were considered statistically significant at a *p*-value < 0.05.

## 3. Results

### 3.1. Age and the EGF/EGFR Pathway in NPC

We analyzed the correlation between age distribution and 186 of 502 patients with NPC who showed recurrence or metastasis. In this study, most patients with NPC were aged between 40 and 60 years (315/502 = 62.7%), whereas few patients were aged ≤ 30 years (23/502 = 4.6%) and ≥70 years (27/502 = 5.4%). However, relatively high recurrence and metastasis rates were observed in these two extreme age groups (56.5% in those aged ≤ 30 years and 48.1% in those aged ≥ 70 years; Figure 1A).

It is still unclear why a higher recurrent/metastatic rate was observed at both ends of the age spectra. We performed a plasma cytokine array to understand whether cytokines contribute to recurrence and metastasis in patients with NPC. Three paired plasma samples were collected from the same patient who initially had a local disease status and then had recurrence or metastasis in the three different age groups: ≤30, 31–69, and ≥70 years. Subsequently, the plasma samples were hybridized with a panel of an array containing 102 anti-cytokine antibodies to identify differentially expressed cytokines in patients’ plasma samples. The cytokine array results indicated that the youngest age group had a stronger plasma EGF signal than did the other two age groups in both primary local disease and recurrence/metastasis status (Figure 1B). EGF is a growth factor ligand that binds to and activates the tyrosine kinase receptor EGFR. We compared the plasma EGF level of patients with recurrent/metastasis by performing ELISA analysis and found that the youngest age group (*n* = 13) had a significantly higher EGF level (average = approximately 310 pg/mL) than did the oldest age group (*n* = 13; approximately 183 pg/mL; * *p* < 0.05; Figure 1C). Because EGF is the major ligand of EGFR, we examined the EGFR protein expression level in NPC tumors through immunohistochemistry (IHC) staining. We observed that 6 of 13 patients in the youngest age group and 1 of 13 patients in the oldest age group exhibited considerably stronger EGFR expression (IHC = grade 3, black bar; Figure 1D–E). These results indicated that the EGF-EGFR pathway might be preferentially activated in young patients with NPC because they have a higher risk of EGF and EGFR overexpression.

### 3.2. EGFR and CDK4/6 Inhibitors Suppressed NPC Cell Growth

EGFR overexpression was frequently detected in 70–80% of NPC tumors and is associated with poor prognosis and outcomes [33]. To block EGF-EGFR signaling, we used the Food and Drug Administration approved EGFR-targeted therapeutics (EGFR inhibitors, EGFRi) in NPC cells, including the monoclonal antibody cetuximab (Erbitux, ERB) and a small molecule afatinib (AFA, a tyrosine kinase inhibitor), both of which can directly bind to EGFR and block the EGFR pathway. The EGFR signaling cascade eventually promotes cell proliferation; thus, one or more cell cycle blockers may also be used to block the EGFR downstream signal. A cell cycle-dependent kinase CDK4/6 inhibitor, palbociclib (PAL), was selected in the current study because we previously reported that this cell cycle inhibitor could suppress NPC tumor growth in an NPC PDX animal model [21]. To test this hypothesis, we first established an NPC cell line derived from our previous genome-sequenced NPC PDX mice models [21] to perform in vitro drug screening tests. This recently established PDX-derived NPC cell line, NPC-B13, was proven to harbor EBV based on the staining of EBV-encoded RNA (Figure 2A). Western blot analysis was performed to evaluate EGFR expression status in the currently available NPC and NPC PDX cell lines. EGFR protein was highly expressed in HK1 and NPC-B13 cells (Figure 2B). Subsequently, we tested the drug sensitivity of a small-molecule drug EGFRi, AFA, in two NPC cells and our newly established NPC-B13 cells. The IC_50_ (IC50 Calculator, AAT Bioquest) range of AFA in HK-1 (EBV-), NPC-B13 (EBV+), and C666-1 (EBV+) cells was 9 nM–2.46 μM, whereas the IC_50_ range of PAL in the same cell lines was 3–35 μM (Figure 2C). Determining the characteristics of (EBV+) NPC cells enables an effective assessment of drug responses.

### 3.3. Two EGFR Inhibitors and a CDK4/6 Inhibitor Suppressed NPC PDX Tumor Growth in an Animal Model

Because EGFR was highly expressed in our NPC PDX cell line, we examined the role of EGFR in NPC tumor-related function. The effects of EGFRi, ERB, AFA, and PAL, a cell cycle blocker, were tested in our PDX-B13 and PDX-ST1 mouse model. As shown in Figure 3A–D, when EGFRi and PAL were applied separately, each of them could inhibit PDX-B13 tumor growth by approximately 30% and 70%, respectively (tumor volume, Figure 3C). EGFRi exerted an additive repressive effect of approximately 90% when combined with PAL. To monitor EGFR expression after different drug treatments, IHC analysis of EGFR was conducted in NPC-PDX-B13 tumors. EGFR protein expression was moderately decreased after AFA or ERB treatment when compared with the control treatment (DMSO; Figure 3F). By contrast, PAL exerted no effect on EGFR expression; however, it blocked approximately 70% of PDX tumor growth. Similar inhibitory effects of combined treatment with EGFRi and PAL were observed on another NPC PDX line, PDX-ST1 (Figure 3G). Although EGFRi and PAL target distinct oncogenic pathways, when applied together, they exerted prominent antitumor effects on the NPC PDX mouse model.

### 3.4. EBV DNA Load in NPC PDX Tumor Correlated with Tumor Weight

Plasma EBV DNA derived from NPC cells is correlated with tumor burden and stage and is a satisfactory prognostic indicator for NPC [26,28]. EBV load in the tumor before and after treatment and in disease progression was unknown. Our NPC PDX mouse model allowed us to monitor changes in the EBV DNA copy in the tumor as well as the tumor size with or without drug treatment. The EBV DNA load in the genomic DNA of PDX-B13 tumor before and after different treatments (from Figure 3) was quantitated using real-time qPCR and normalized using beta-actin as control. The EBV DNA load in PDX tumors tended to positively correlate with tumor weight (Figure 4). These results indicated that EBV DNA copies/cells might reflect the size of an NPC tumor and also signify whether the drug treatment would have a favorable outcome.

### 3.5. EGFR Inhibitors and CDK4/6 Inhibitor Blocked EGFR Phosphorylation

To confirm whether EGFRi and PAL can suppress EGFR signaling and the cell cycle pathway, respectively, protein lysates of drug-treated NPC-PDX-B13 tumors were subjected to Western blot analysis. Activation of the EGFR pathway was examined based on the phosphorylation of the EGFR (EGFR-*p*) molecule; however, treatment of PDX-B13 tumor with PAL, AFA, ERB, PAL+AFA, and PAL+ERB could reduce the EGFR phosphorylation level when compared with the negative control (DMSO), indicating that all the tested drugs could suppress the EGFR pathway by 70–90% (Figure 5). Thus, when applied alone, two EGFR inhibitors, ERB and AFA, and PAL could inhibit PDX tumor growth. However, the combination of PAL with either one of these EGFR inhibitors exerted a potent effect on the EGFR pathway and resulted in cell cycle suppression. These results demonstrate cross-talk between EGFR and cell cycle pathways that merged downstream to promote cell growth. Hence, a combination of inhibitors blocking both pathways could lead to a significant tumor growth reduction together with a decrease in proliferation indexes, including RB1 phosphorylation, E2F1, CCND1, and PCNA expression levels, in the NPC PDX model (Figure 5).

### 3.6. Transcriptomic Profile of Drug-Treated NPC PDX Tumors

To analyze the effectiveness of drug treatment, the RNA sequencing of control and drug-treated NPC PDX tumors was performed. In summary, the mean number of aligned reads was 63.3 million (range = 59.7–66.4). On average, 98.4% of reads mapped to the human genome were (range = 96.6–99.0). We conducted differential expression (DE) analysis of the xenograft by using three analysis designs. In design A, we compared the DE of tumors treated with EGFRi versus without EGFRi (AFA and ERB versus control, PAL). The log-ratio (M) and mean normalized count (A), and MA plot results are shown in Figure 6A. In design A, only two genes (CCL5 and FYB) were significantly downregulated (adjusted *p* < 0.05, as shown in the MA plot; Figure 6A). In design B, we compared the DE of tumors treated with or without PAL (PAL, PAL + AFA, and PAL+ERB versus control, AFA, and ERB). A total of 339 genes exhibited an absolute log2 fold change larger than 1 and an adjusted *p*-value smaller than 0.05. To analyze significant pathways involved after drug treatment, GO enrichment analysis using 339 genes was performed; the results are shown in Figure 6B. Upregulated genes were associated with a cellular response to stimuli, and downregulated genes were associated with mitotic activity and cell cycle processes. In design C, we compared the DE of tumors treated with a combination of inhibitors CDK4/6i and EGFRi versus a single agent ((PAL + AFA and PAL + ERB) versus (PAL, AFA, and ERB)). A total of 271 genes exhibited an absolute log2 fold change larger than 1 and an adjusted *p*-value smaller than 0.05. The results of GO enrichment analysis are shown in Figure 6C. In both B and C designs, we found that the majority of downregulated genes were associated with cell cycle processes, indicating that PAL can block the cell cycle and inhibit tumor proliferation.

### 3.7. EGFR Is a Prognostic Factor for and Therapeutic Target against NPC

EGFR was reported to be highly expressed in the primary tumor site of NPC, and it is a poor prognosis factor [33]. We examined the role of EGFR in local recurrence and metastatic tissues of NPC. EGFR expression was analyzed through IHC in 42 tissues related to local recurrence and 90 metastatic tissues of NPC obtained between 2005 and 2019 from the tissue bank of CGMH, Linkou, Taiwan. EGFR was found to be highly expressed in local recurrent tumors (≥grade 2, 32/42, 76.2%) and metastatic tumors (≥grade 2, 64/90, 71.1%; Figure 7A,B). Patients with metastatic NPC whose EGFR expression in the tumor was lower than grade 3 had significantly better survival outcomes than did those with grade 3 expression (*p* = 0.008; Figure 7C). However, the percentage of EGFR-positive cells in tumors did not affect overall survival (Figure 7D).

One patient with NPC who received induction chemotherapy combined with an immune checkpoint inhibitor combined with concurrent chemoradiotherapy for local residual disease and UFT later developed lung metastasis with high EGFR expression (Figure 7E). As salvage therapy, the patient was further treated with two courses of PAL combined with AFA. Although the patient showed maximal grade 2 myelosuppression and a grade 2 skin rash, a decrease in plasma EBV DNA load from approximately 5000 copies/mL to approximately 1000 copies/mL was observed, implying some responsiveness to treatment drugs (Figure 7F). A follow-up chest X-ray exam revealed a stable disease status (Figure 7G). Our data suggest that PAL + AFA may serve as a salvage treatment for NPC.

## 4. Discussion

### 4.1. EGF-EGFR Pathway Activation in the Young Age Group

Compared with patients with NPC aged >30 years, those aged ≤ 30 years have been reported to have a more advanced stage and poor pathologic differentiation in initial diagnosis but more favorable outcomes, possibly due to tolerance to more aggressive chemotherapy and better family and financial support [7,34]. After intensity modulation radiation therapy (IMRT) treatment, juvenile patients exhibited better outcomes but more severe cytotoxicity than did adults [35]. Compared with adult patients with NPC, juvenile patients with NPC demonstrate specific features, including a higher level of C-kit and EBV LMP1 expression and a lower level of p53 and Bcl2 in the tumor and anti-EBV IgG and IgA in peripheral blood [36,37]. In the current study, the results of the cytokine array study revealed that the plasma EGF level was higher in the young age group than in the old age of patients with NPC with local recurrence or metastasis; in addition, tumor EGFR expression was higher in the young age group. EGF could induce epithelial-mesenchymal transition in NPC and HK-1 cells through microRNA-related feedback regulation [38]. These results indicate that the EGF–EGFR signaling pathway may play a role in tumor progression in young patients with NPC.

### 4.2. EGFR Is a Prognostic Factor for Metastatic NPC

In a meta-analysis on NPC, EGFR overexpression was reported to be a poor prognostic factor for overall survival, disease-free survival, and locoregional control but not for distant metastasis-free survival [33]. In the current study, EGFR overexpression (grade 2 and 3) was detected in approximately 70% of patients with recurrence or metastasis, and high expression was noted in those with local recurrence (38%, grade 3) and distant metastasis (41%, grade 3). Furthermore, the overall survival duration was significantly shorter in patients with metastatic NPC with EGFR overexpression (grade 3; *p* = 0.008 **; Figure 7).

### 4.3. EGFR Is a Target in NPC Management

Because EGFR is a poor prognostic factor, EGFRi has been used in NPC management. ERB, an EGFR monoclonal antibody, demonstrated effectiveness in combination therapy with cisplatin and IMRT in locally advanced NPC in a phase II trial [39]. In metastatic NPC, cisplatin-refractory patients with NPC who received ERB as combination therapy with carboplatin had a 12% response rate, and 48% had a stable disease status [40].

Another group of EGFR inhibitors, tyrosine kinase inhibitors including gefitinib and erlotinib, failed to show promising results in patients with recurrent/metastatic NPC [41]. However, a new generation of EGFRi, AFA, which is an irreversible EGFR/HER2 inhibitor, was found to inhibit cell growth, augment the anticancer effect of gemcitabine [42], and enhance radiosensitivity in NPC [43]. In the current study, we showed that either ERB or AFA could inhibit NPC PDX tumor growth. Furthermore, RNA sequencing comparing EGFRi treatment and baseline tumor showed that two genes, chemokine (C-C motif) ligand 5 (CCL5) and FYB, were significantly downregulated (Figure 6). CCL5 can be a potential biomarker and therapeutic target in NPC based on a cancer cell line secretome profiling study [44]. EBV-infected NPC cells showed increases CCL5 expression and enhanced tumor angiogenesis [45]. FYB gene encoding adhesion and degranulation adaptor protein (ADAP), which is expressed on platelets and many inflammatory cells, is related to cell motility, proliferation, activation, and cytokine production [46]. The relationship of EGFRi with these two genes in NPC should be studied in future studies.

### 4.4. Combination Therapy of EGFRi + CDK4/6i Exerted an Additive Effect on NPC PDX Tumor

CDK4/6i is a potent anticancer drug and a radiosensitizer in NPC [21,47,48]. Recently, combination therapy of CDK4/6i with hormone therapy was used to treat hormone receptor-positive breast cancer [49]. In NPC, CDK4/6i in co-treatment with alpelisib (PI3K inhibitor) [47], gemcitabine (antimetabolite) [21], and SAHA (HDAC inhibitor) [50] exerted antitumor effects on a preclinical animal model. In the current study, an EGFRi, either ERB or AFA, could augment the antitumor effect of PAL in an NPC PDX model. In HPV-unrelated, cisplatin-resistant, ERB-resistant, recurrent, and metastatic head and neck squamous cell carcinoma (HNSCC), combination therapy of PAL and ERB demonstrated some promising results in a phase II clinical trial [51]. A follow-up study of the aforementioned clinical trial in HNSCC reported that co-treatment of PAL and ERB tended to prolong median overall survival compared with ERB alone [52]. These results may encourage the application of PAL together with EGFRi for the treatment of NPC.

## 5. Conclusions

EGF-EGFR pathway was activated in young age group of NPC patients and EGFR overexpression was frequently found in recurrent/metastatic NPC patients. Targeting in EGFR via monoclonal antibody or tyrosine kinase inhibitor could decrease xonograft growth in pre-clinical animal model. Further combining EGFRi with CDK4/6 inhibitor had additive effect on decreasing tumor growth in animal model. These results may provide base for further clinical trial in NPC management.

## Figures and Tables

**Figure 1 cancers-13-02954-f001:**
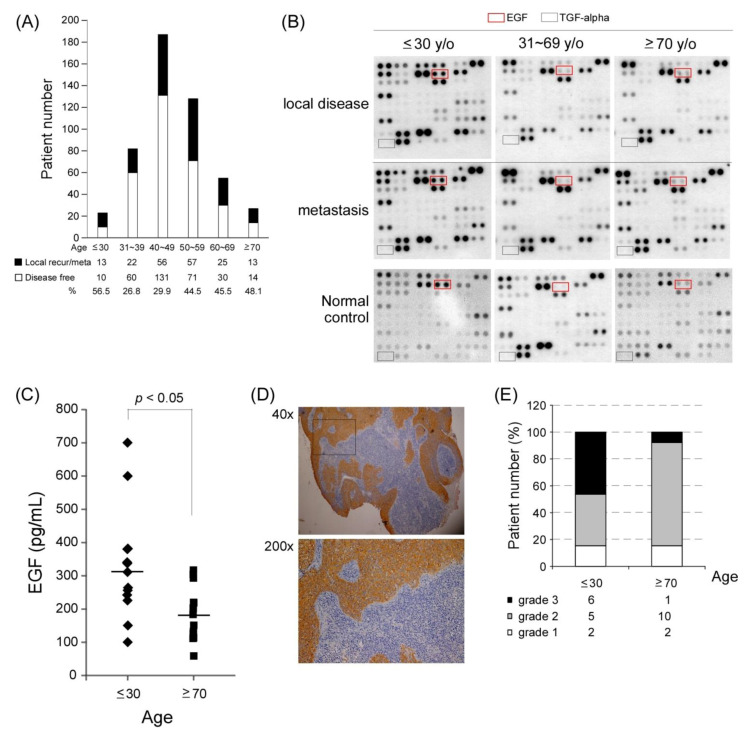
Various age groups of patients with NPC and the EGFEGFR pathway. (**A**) Patient distribution according to age groups. Patients aged ≤ 30 and ≥70 years had a higher rate of local recurrence and metastasis (black bar) when compared with other age groups. (**B**) Findings of the plasma cytokine array in different age groups of patients with metastatic NPC. Epithelial growth factor receptor (EGFR) ligands, including EGF and TGF-alpha, were screened in different age groups with local disease or distant metastasis. (**C**) The mean plasma EGF levels of patients with recurrent or metastatic NPC were 310.9 pg/mL in those aged ≤ 30 years (*n* = 13) and 183.2 pg/mL in those aged ≥ 70 years (*n* = 13; *t*-test, *p* < 0.05). (**D**) IHC staining results of EGFR expression (grade 3) in one NPC tumor tissue. (**E**) Tissue EGFR expression in patients with NPC aged ≤ 30 (*n* = 13) and aged ≥ 70 years (*n* = 13; grade 3, black bar; grade 2, grey; and grade 1, white).

**Figure 2 cancers-13-02954-f002:**
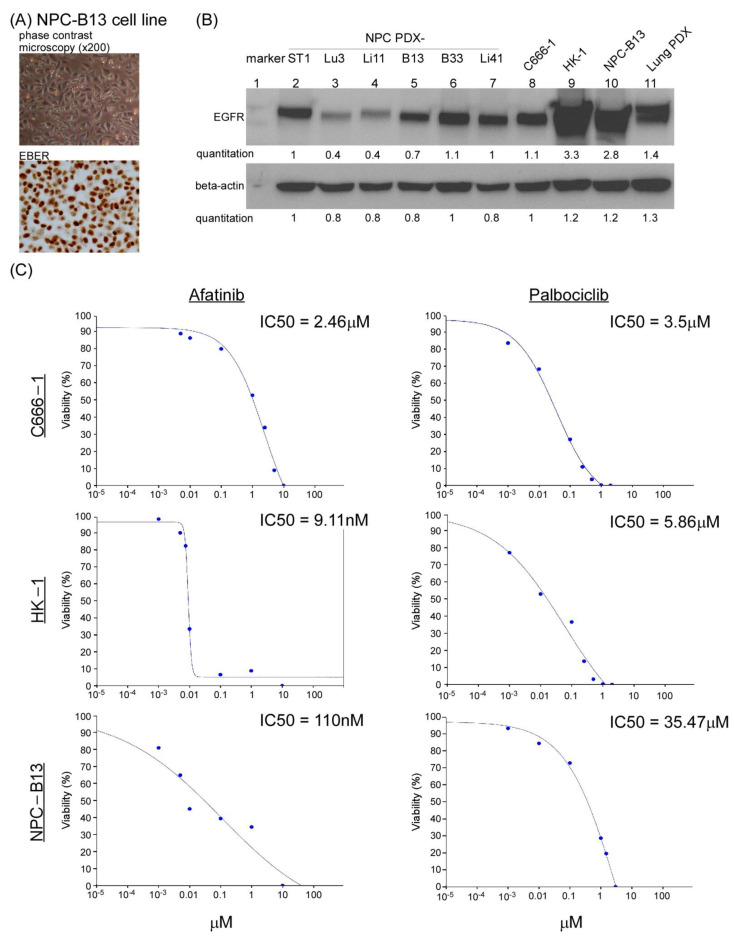
NPC PDX tumor-derived cell line and in vitro drug tests. (**A**) Morphology of NPC-B13 cells, a newly established EBV (+) cells derived from NPC PDX-B13 (upper panel) and its EBV encoded RNA (EBER) staining (lower panel). (**B**) Western blot analysis of EGFR expression in six NPC PDX tumors (lanes 2–7) and three NPC cells (lanes 8–10) and lung cancer PDX (lane 11, EGFR-positive control). The uncropped Western blots have been shown in Appendix A. The HPV genotyping was confirmed in Appendix A. The charcteristics of patients for the NPC PDX were listed in Appendix A. (**C**) IC_50_ of afatinib and palbociclib in NPC cell lines, C666-1, HK-1, and NPC-B13.

**Figure 3 cancers-13-02954-f003:**
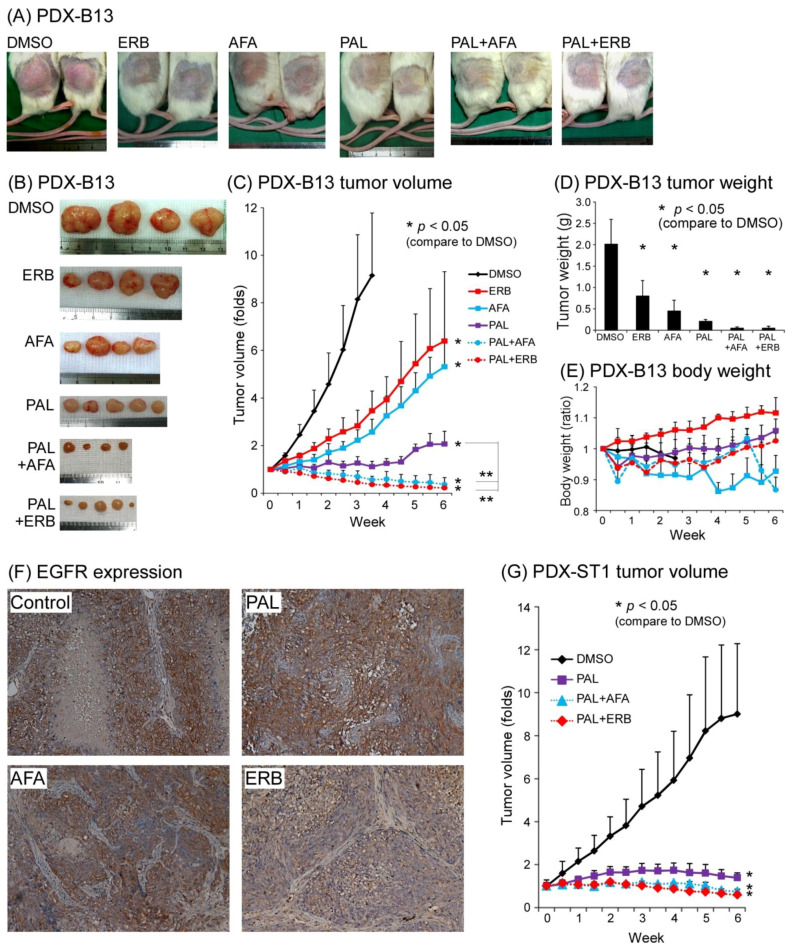
NPC PDX drug screening. PDX-B13, NPC PDX-B13F11; PDX-ST1, NPC PDX-ST1F15. (**A**,**B**) Mouse gross tumor, (**C**) tumor volume, (**D**) tumor weight, and (**E**) body weight change. (**F**) EGFR expression before (Control) and after treatment (PAL, AFA, and ERB) (200×) (**G**) NPC PDX-ST1 drug screening. ** *p* < 0.01. Abbreviations: AFA: afatinib; ERB: erbitux; PAL: palbociclib.

**Figure 4 cancers-13-02954-f004:**
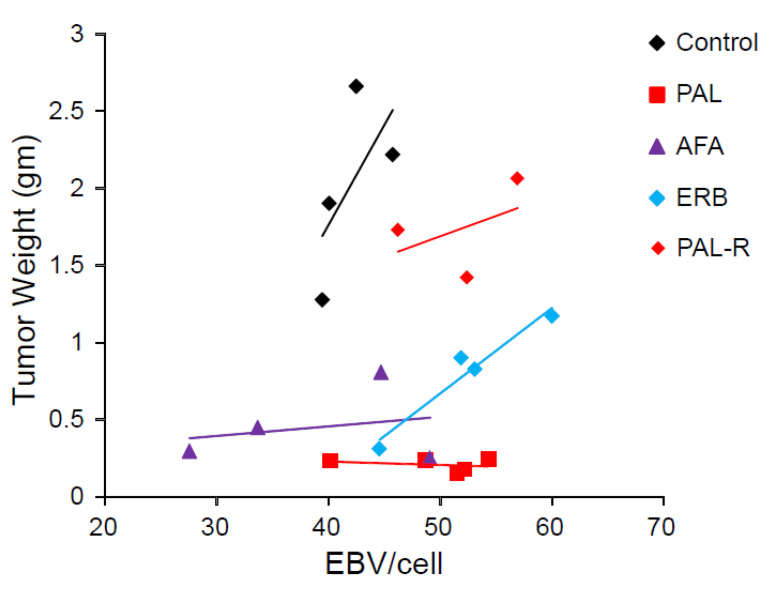
EBV copies per cell related to different treatments of NPC PDX. Abbreviations: PAL, palbociclib; AFA, afatinib; ERB, erbitux; PAL-R, pablociclib resistance.

**Figure 5 cancers-13-02954-f005:**
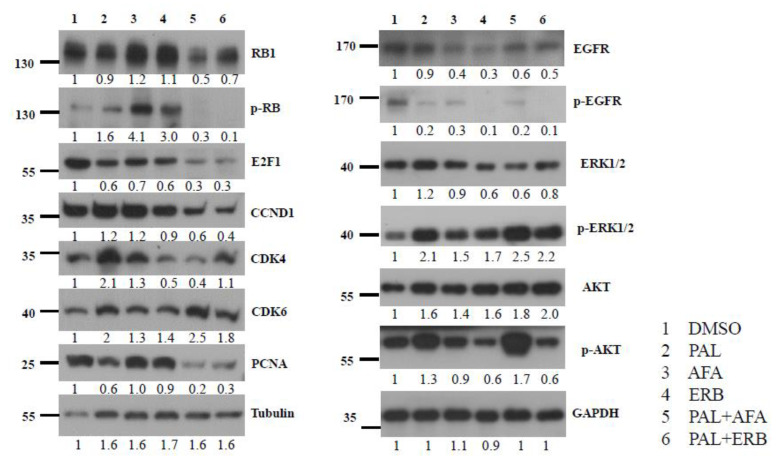
Western blot analysis of key protein molecules of EGFR and cell cycle pathways in NPC PDX-B13 tumors with or without EGFRi and/or palbociclib treatment.

**Figure 6 cancers-13-02954-f006:**
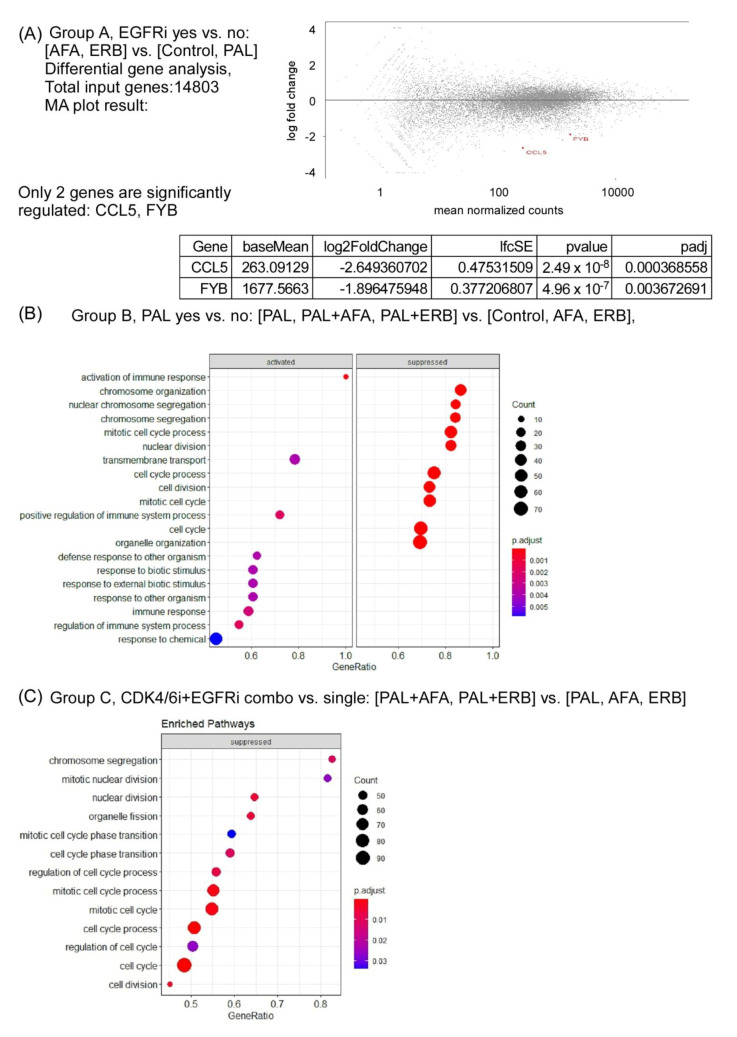
Bulk RNA sequencing of NPC PDX with or without EGFRi and/or CDK4/6i treatment. (**A**) Group A, EGFRi yes vs. no: (AFA and ERB0 vs. (Control and PAL); (**B**) group B, PAL yes vs. no: (PAL, PAL + AFA, and PAL + ERB) vs. (Control, AFA, and ERB); (**C**) group C, CDK4/6i+EGFRi combo vs. single: (PAL + AFA and PAL + ERB) vs. (PAL, AFA, and ERB).

**Figure 7 cancers-13-02954-f007:**
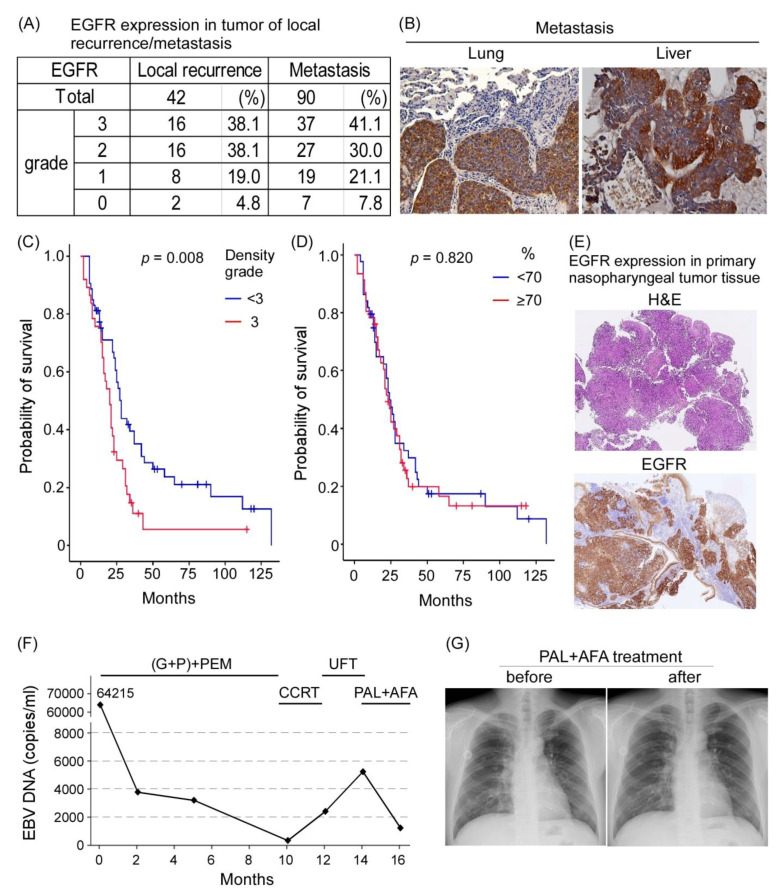
EGFR in patients with clinical local recurrence of NPC or metastatic NPC. (**A**) EGFR expression in patients with local recurrence and distant metastasis. (**B**) IHC staining of EGFR in local recurrence and distant metastasis tumors. (**C**) Probability of patient survival between EGFR expression density grade < 3 vs. 3 showed statistical significance (*p* = 0.008); and (**D**) probability of survival between EGFR expression percentage grade < 3 vs. 3 had no statistical significance (*p* = 0.82). (**E**) EGFR expression in the primary tumors of patients with NPC who received salvage therapy. (200×) (**F**) Plasma EBV DNA load reflecting that the clinical treatment response of the patient was decreased after PAL + AFA treatment. (**G**) Chest X-ray exam before and after PAL + AFA treatment at 2-month intervals revealed stable disease status.

## Data Availability

Data is contained within the article or Appendix A. The data presented in this study are available in this article and Appendix A.

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
