# Peer review of "Combination of Epithelial Growth Factor Receptor Blockers and CDK4/6 Inhibitor for Nasopharyngeal Carcinoma Treatment"

_cancers, 2021, doi:10.3390/cancers13122954_

Round 1

Reviewer 1 Report

No additional comments.

Reviewer 2 Report

The authors have addressed all my concerns and therefore I support publication without further changes.

Reviewer 3 Report

Authors have included all the suggestions. Manuscript looks impactful. 

This manuscript is a resubmission of an earlier submission. The following is a list of the peer review reports and author responses from that submission.

Round 1

Reviewer 1 Report

The study represents analysis of response of NPC to combined inhibitors of EGR and CDK4/6 complex.

This is an ex-vivo study if NPC primary tumor and cell lines.

The main concerns in this study are:

  • The number of PDX and the cell lines are too small to address the demographic and pathologic heterogeneity of this entity. To draw any conclusion on the effectiveness of treatment multiple PDX and cell lines representing a spectrum of these patients.
  • The expression data and illustrations of the markers and tumor size do not show any significant changes. Tumor size illustration in particular are questionable due to the different magnification used.

Author Response

Comments and Suggestions for Authors

  1. The introduction was poorly re-written. Not sure why the authors discussed colon and breast cancer? It seems to be open-ended statement. This section needs to re-write to reflect the overall purpose of this study.

=>Answer: Thanks reivewer’s suggestion. We had re-written the introduction focusing on NPC as in the revised manuscript.

  1.    Figure 1 B. What age group was used for ‘Normal Control’? It should include all control for <=30, 31-69 and >=70 years of age.

=>Answer: We had added healthy persons’ samples from 3 different age group as in the revised Fig. 1B. In age ≤30 group had higher EGF concentration in plasma including patients and healthy control than other age groups. The result is reasonable that young age group had higher constitutional growth factor(s) background with vigorous performance status which may be taken advantage by cancer cell than other age group.

  1.    Figure 1. Legend ‘…very strong expression (grade 3) was higher in age <=30 (6/13) than age >=??? (1/13), missing 70

=>Answer: Thanks for correction. It had been corrected in revised manuscript.

  1.    It is not clear on the purpose of Figure 1 D, and E.  Authors need to explain why the younger age group had higher EGF levels than other age groups. How do these findings impact the overall treatment plans?

=>Answer: From our clinical observation, young age (≤30) and old age (≥70) NPC patients had higher recurrent rate as shown in Fig 1A. We tried to figure out any factor(s) to give rise to this result. Through cytokine array analysis from different age group, we found EGF was higher in young age group. This result was reasonable that young patient had higher growth factor background with energetic activity than old age group that cancer could take advantage of this condition. EGF needed to go through EGFR to perform its influence on cell activity including proliferation. So we checked the EGFR expression status in these two different age group and found young age group had higher EGF-EGFR expression than old as shown in Fig 1D & 1E. Since EGFRi had been discovered and prescribed in some cancer treatment, NPC patients with high plasma EGF and/or high tumor EGFR expression may be benefited from EGFRi treatment.

  1.    HPV may be the etiologic factor in some EBV-negative, non-keratinizing NPCs
             The authors should determine the HPV status on all cell lines used in this study.

=>Answer: Since C666-1 was shown harboring EBV [Int J Cancer 1999 Sep 24;83(1):121-6.] and NPC-B13 was EBV-positive as shown in Fig 2A, we checked the HPV status in HK-1 cell line. Using HPV Blot Chip for HPV genotyping [Int J Cancer. 2021 Feb 1;148(3):665-672.], HK-1 didn’t have endogenous HPV and positive control of oropharyngeal cancer PDX showed positive for HPV-18 as shown in supplement data 1.

  1.    Figure 2. B Authors should explain why the expression of EGFR in links 3 and 4 significantly lower the rest. Quantitate analysis of the western blot is needed. 

=>Answer: We didn’t know why the expression of EGFR was significant lower in lane 3 and 4 (NPC PDX-Lu3 [64y/o] and -Li11 [52y/o]). Probably due to patient variation and tumor heterogeneity. We redid and quantitated the Western blot of the PDXs and cell lines as shown in revised Fig 2B. We had renamed our NPC PDX system as following: NPC PDX-(tissue source abbreviation)(patient number). The nomenclature of the NPC PDX-ST1 and PDX-B13 used in current manuscript is corresponding to NPC PDX-ST (#1) and NPC PDX-Bone (#13) in reference [J Exp Clin Cancer Res. 2018 Sep 20;37(1):233.].

  1.    Figure 2.C Fitting all the dose-response curves to find the absolute IC50 is needed. Did the dose-response curve only been done once? n=??? Please provide all the data files.

=>Answer: We had checked the IC50 at least twice and the results had been shown in the supplement data----. The initial data was calculated from measured wavelength 590nm and we had revised the calculated difference between measurement (590nm) and reference measurement (630nm) in the revised manuscript. We had recalculated the IC50 via IC50 Calculator/AAT Bioquest and the data was in supplement data-2.

  1.    Figure 5. It is difficult to judge without a Quantitative Analysis of the western blots. It is not clear why pEGFR was so low even with Palbo alone? Did the western blots been repeated more than 3 times? (n=???) Authors should include pStat 5 and pPLC-γ1 status for the activation of EGFR signaling.

=>Answer: Please refer the updated quantified Western blots (TIF file). We basically repeat the Western blot twice (n=2), however, for the two-drug combination treated tumors, in particular, sample #5. PAL+AFA and #6. PAL+ERB, due to the effective drug treatment the tumor size of these 2 samples were rather small (2~5mm diameter, Fig.3B). In addition, we also used half of these tumors to perform RNA seq, therefore, in fact we have already used up all the samples (#5, #6). Besides, we have tried different dilution conditions and different antibodies from different companies, and not all of the western blot analyses and antibodies were successful and usable. Hence, we are very sorry that in the mean time we are not able to perform any new western blot experiment.

  1.    Figure. 6. Not until high-resolution images are included. It is hard to analyze the data.

=>Ans: We had upload high-resolution images of Fig. 6 in revised manuscript.

Reviewer 2 Report

1.    The introduction was poorly re-written. Not sure why the authors discussed colon and breast cancer? It seems to be open-ended statement. This section needs to re-write to reflect the overall purpose of this study.
2.    Figure 1 B. What age group was used for ‘Normal Control’? It should include all control for <=30, 31-69 and >=70 years of age.
3.    Figure 1. Legend ‘…very strong expression (grade 3) was higher in age <=30 (6/13) than age >=??? (1/13), missing 70
4.    It is not clear on the purpose of Figure 1 D, and E.  Authors need to explain why the younger age group had higher EGF levels than other age groups. How do these findings impact the overall treatment plans?
5.    HPV may be the etiologic factor in some EBV-negative, non-keratinizing NPCs
           The authors should determine the HPV status on all cell lines used in this study.
6.    Figure 2. B Authors should explain why the expression of EGFR in links 3 and 4 significantly lower the rest. Quantitate analysis of the western blot is needed. 
7.    Figure 2.C Fitting all the dose-response curves to find the absolute IC50 is needed. Did the dose-response curve only been done once? n=??? Please provide all the data files.
8.    Figure 5. It is difficult to judge without a Quantitative Analysis of the western blots. It is not clear why pEGFR was so low even with Palbo alone? Did the western blots been repeated more than 3 times? (n=???) Authors should include pStat 5 and pPLC-γ1 status for the activation of EGFR signaling.
9.    Figure. 6. Not until high-resolution images are included. It is hard to analyze the data.

Author Response

Reviewer 2

Comments and Suggestions for Authors

Li et al. have shown the effect of combination of epithelial growth factor receptor blockers and CDK4/6 inhibitor in NPC treatment. In conclusion, authors demonstrated the application of PAL together with EGFRi in NPC treatment. Results are nicely and clearly written. Figures are well made. Study design is sound. References are up to date. I have following comments. 

1. Please mention full form of PDX-B13 and PDX-ST in the abstract section.

=>Answer: We had renamed our NPC PDX system as following:

NPC PDX-(tissue source abbreviation)(patient number). The nomenclature of the NPC PDX-ST1 and PDX-B13 is corresponding to NPC PDX-ST (#1) and NPC PDX-Bone (#13) in reference [J Exp Clin Cancer Res. 2018 Sep 20;37(1):233.]. Due to the regulation of “Cancers”, we can’t sign the reference in the abstract. The abstract should be a total of about 200 words maximum. We had described in “Materials and methods” section 2.6 and 2.7 of revised manuscript.

2. In the introduction section, paragraph 1, I would highly suggest to include information related to NPC only. Also it would be better if authors will include more clinical information regarding NPC.

=>Answer: Thanks for Reviewer’s suggestion. We had re-written the introduction focusing on NPC as in the revised manuscript.

3. In the same section, therapy resistance should also be mentioned. 

=>Answer: We included some mechanism proposed in therapy resistance in introduction as in the revised manuscript.

4. Introduction section, paragraph 2, please cite the references related to EGFR in NPC and include some background information related to EGFR in NPC.

=>Answer: In our revised manuscript, EGFR related to NPC was reviewed and EGRFi in NPC was also reviewed.

5. Method section needs more elaboration. Patient demographic profile would be better to include in suppl. table. Xenograft section should be written in detail.

=>Answer: Patient demographic profile had been added in revised manuscript supplement Table 1 (supplement data-3). The nomenclature system of our NPC PDX was renewed in the revised manuscript as following: NPC PDX-(tissue source abbreviation)(patient number). The NPC PDX-ST1 and PDX-B13 corresponded to NPC PDX-ST and PDX-Bone in reference [J Exp Clin Cancer Res. 2018 Sep 20;37(1):233.]. NPC PDX-B33 and NPC PDX-Li41 were new established NPC PDX lines and were under characterization. NPC PDX-B13 had paired EBV-positive cell line, NPC-B13. NPC-B13 cell line had been passaged for more than 80 passages with EBV episome 40~50 copies/cell in vitro but would not form xenograft in NOD/SCID mice. All these PDX lines harbored EBV except NPC PDX-Li41.

6. Figure 2B. It would be better to include representative clean blots of EGFR and actin along with densitometric analysis.

=>Answer: We redid the Western blots of EGFR expression with quantitation in revised manuscript Fig. 2B.

7. Figure 5, please label conditions on top of the blots. Also, it would be better if author can show nicely cropped blots with space in all edges. Blots are very tightly cropped. 

=>Answer: Please refer the updated quantified Western blots (Fig. 5, TIF file). For the two-drug combination treated tumors, in particular, sample #5. Pal+Afa and #6. Pal+Erb, due to the effective drug treatment the tumor size of these 2 samples were rather small (2~5mm diameter, Fig.3B). In addition, we also used half of these tumors to perform RNA seq and RT-PCR, therefore, we only have limited amount of samples for Western blot analysis. In order to save protein samples, after protein transfer the nitrocellulose blot was further divided/ cut into 2 pieces (corresponding to high and low molecular weight protein size) and these 2 smaller pieces of the blot were used separately to hybridize with two different antibodies. Sometimes we even divided the blot into 3 smaller pieces to hybridize with 3 different antibodies. Therefore “the blots were tightly cropped” because we “cropped/cut” the blot before Western blot hybridization not after X-ray film development to save limited amount of protein samples from tumor #5 and #6. We included the Western blot results of p-AKT and internal control GAPDH.

8. For fig 6B and C, please use high resolution images. 

=>Answer: We had upload high-resolution images of Fig. 6 in revised manuscript.

Reviewer 3 Report

Li et al. have shown the effect of combination of epithelial growth factor receptor blockers and CDK4/6 inhibitor in NPC treatment. In conclusion, authors demonstrated the application of PAL together with EGFRi in NPC treatment. Results are nicely and clearly written. Figures are well made. Study design is sound. References are up to date. I have following comments. 

  1. Please mention full form of PDX-B13 and PDX-ST in the abstract section.
  2. In the introduction section, paragraph 1, I would highly suggest to include information related to NPC only. Also it would be better if authors will include more clinical information regarding NPC.
  3. In the same section, therapy resistance should also be mentioned. 
  4. Introduction section, paragraph 2, please cite the references related to EGFR in NPC and include some background information related to EGFR in NPC.
  5. Method section needs more elaboration. Patient demographic profile would be better to include in suppl. table. Xenograft section should be written in detail.
  6. Figure 2B. It would be better to include representative clean blots of EGFR and actin along with densitometric analysis.
  7. Figure 5, please label conditions on top of the blots. Also, it would be better if author can show nicely cropped blots with space in all edges. Blots are very tightly cropped. 
  8. For fig 6B and C, please use high resolution images. 

Author Response

The study represents analysis of response of NPC to combined inhibitors of EGR and CDK4/6 complex. This is an ex-vivo study if NPC primary tumor and cell lines.

The main concerns in this study are:

1. The number of PDX and the cell lines are too small to address the demographic and pathologic heterogeneity of this entity. To draw any conclusion on the effectiveness of treatment multiple PDX and cell lines representing a spectrum of these patients.

=>Answer:” The number of PDX and the cell lines are too small”.

We had used two NPC PDX lines harboring endogenous EBV (NPC PDX-ST[#1] and PDX-Bone [#13] ) to demonstrate the efficacy of combination of EGFRi and CDK4/6i in animal model as shown in Fig.3A-E (PDX-Bone) and Fig.3G (PDX-ST). Three NPC cell lines including 2 harboring EBV, C666-1 and newly established NPC-B13, and one EBV-negative NPC cell line, HK-1, were tested the EGFR protein expression and IC50 of EGFRi and CDK4/6i as shown in Fig.2. The material number used was adequate as comparing manuscript published in “Cancers” journal.

Establishment and Characterization of Humanized Mouse NPC-PDX Model for Testing Immunotherapy

Cancers (Basel) 2020 Apr 22;12(4):1025.doi: 10.3390/cancers12041025

-------one NPC PDX line.

2. The expression data and illustrations of the markers and tumor size do not show any significant changes. Tumor size illustration in particular are questionable due to the different magnification used

=>Ans: “Tumor size illustration in particular are questionable due to the different magnification used”.

Please note that there was a ruler scale (cm) below each group of tumors (Fig.3B). The ruler scale in different group was in fact the same. Therefore, tumor size from different groups can be compared.
